# Naturally Occurring Chalcones with Aggregation-Induced Emission Enhancement Characteristics

**DOI:** 10.3390/molecules28083412

**Published:** 2023-04-12

**Authors:** Iwona Budziak-Wieczorek, Daniel Kamiński, Alicja Skrzypek, Anna Ciołek, Tomasz Skrzypek, Ewa Janik-Zabrotowicz, Marta Arczewska

**Affiliations:** 1Department of Chemistry, University of Life Sciences in Lublin, Akademicka 15, 20-950 Lublin, Poland; alicja.skrzypek@up.lublin.pl (A.S.); anna.ciolek@up.lublin.pl (A.C.); 2Institute of Chemical Sciences, Maria Curie-Skłodowska University, Pl. Marii Curie-Sklodowskiej 3, 20-031 Lublin, Poland; daniel.kaminski@umcs.pl; 3Department of Biomedicine and Environmental Research, Faculty of Medicine, The John Paul II Catholic University of Lublin, Konstantynów 1J, 20-708 Lublin, Poland; tomasz.skrzypek@kul.pl; 4Department of Cell Biology, Institute of Biological Sciences, Maria Curie-Sklodowska University, ul. Akademicka 19, 20-033 Lublin, Poland; 5Department of Biophysics, University of Life Sciences in Lublin, Akademicka 13, 20-950 Lublin, Poland

**Keywords:** AIEE, acetylcholinesterase (AChE), aggregation, butyrylcholinesterase (BuChE), natural chalcones

## Abstract

In this paper, the natural chalcones: 2′-hydroxy-4,4′,6′-trimethoxychalcone (HCH), cardamonin (CA), xanthohumol (XN), isobavachalcone (IBC) and licochalcone A (LIC) are studied using spectroscopic techniques such as UV–vis, fluorescence spectroscopy, scanning electron microscopy (SEM) and single-crystal X-ray diffraction (XRD). For the first time, the spectroscopic and structural features of naturally occurring chalcones with varying numbers and positions of hydroxyl groups in rings A and B were investigated to prove the presence of the aggregation-induced emission enhancement (AIEE) effect. The fluorescence studies were carried out in the aggregate form in a solution and in a solid state. As to the results of spectroscopic analyses conducted in the solvent media, the selected mixtures (CH_3_OH:H_2_O and CH_3_OH:ethylene glycol), as well as the fluorescence quantum yield (ϕ_F_) and SEM, confirmed that two of the tested chalcones (CA and HCH) exhibited effective AIEE behaviour. On the other hand, LIC showed a large fluorescence quantum yield and Stokes shift in the polar solvents and in the solid state. Moreover, all studied compounds were tested for their promising antioxidant activities via the utilisation of 1,1- diphenyl-2-picrylhydrazyl as a free-radical scavenging reagent as well as potential anti-neurodegenerative agents via their ability to act as acetylcholinesterase (AChE) and butyrylcholinesterase (BuChE) inhibitors. Finally, the results demonstrated that licochalcone A, with the most desirable emission properties, showed the most effective antioxidant (DPPH IC_50_ 29%) and neuroprotective properties (AChE IC_50_ 23.41 ± 0.02 μM, BuChE IC_50_ 42.28 ± 0.06 μM). The substitution pattern and the biological assay findings establish some relation between photophysical properties and biological activity that might apply in designing AIEE molecules with the specified characteristics for biological application.

## 1. Introduction

Organic luminophores with high quantum efficiencies have attracted attention due to their widespread applications in organic optoelectronics [1,2,3], laser displays [4,5], light-emitting diodes (OLEDs) [6], fluorescent sensors and biological imaging [7,8,9,10]. However, many traditional fluorophores usually show strong fluorescence only in dilute solutions. In the aggregated and solid states, they tend to be weakly emissive or even non-emissive, which is attributed to the strong intermolecular π-π stacking interactions that absorb excitation energy [7,11,12]. Preventing this problematic ACQ (aggregation-caused quenching) effect associated with the loss of emission efficiency is the key to obtaining solid organic fluorophores with improved optical properties.

In response to this demand, new phenomena have been discovered in the last decade involving aggregation-induced enhanced emission (AIEE) and aggregation-induced emission (AIE), which are the opposite of ACQ. In these cases, the fluorophores become highly fluorescent in the condensed state due to the restriction of their intramolecular motions (RIMs), including both the restriction of intramolecular rotations (RIRs) and the restriction of intramolecular vibrations (RIVs), preventing non-radiative relaxation [7,11,13,14,15,16]. Moreover, many other mechanisms have been proposed to explain the AIE phenomena, such as J-aggregate formation, twisted intramolecular charge transfer (TICT), excited-state intramolecular proton transfer (ESIPT) and Förster resonance energy transfer (FRET) [7,11,13,17].

The AIE phenomenon has opened up tremendous opportunities in various fields, including biological assays, materials science, and photonics [16,18,19]. More importantly, the AIE concept has yielded remarkable prospects in bioimaging applications as sensitive fluorescent biosensors for the detection of functional biomacromolecules (e.g., DNA [20]) and even for a significant increase in the sensitivity of immunoassays [21]. In particular, the development of AIE in the biomedical field has shown inexhaustible momentum, having a great potential for health condition monitoring at the molecular and cellular levels, including real-time monitoring of the progress of cancer therapy [22]. The AIE/AIEE effect is observed in a variety of small organic π-conjugated molecules with donor-π-acceptor (D-π-A) structures, but these mostly involve newly formed structures or the derivatives of existing molecules due to the synthesis. Despite intensified attempts to discover natural molecules characterised by AIE properties for biomedical applications, the number of such natural fluorophores still needs to be increased.

Researchers’ attention has been attracted to chalcones as AIE properties have been reported for several of them [9,23,24,25]. Chalcones are a class of compounds with a characteristic scaffold consisting of two substituted aromatic rings connected by the carbonyl group and the α,β-unsaturated system (aromatic ketone and enone), giving them unique chemical and biological properties [26,27]. Naturally occurring chalcones are a subclass of the compounds belonging to the flavonoid family, and they are found in many edible plants, fruits and vegetables [28,29] that exhibit a wide range of bioactivity, including antioxidant, anti-inflammatory, antimicrobial and anticancer properties [30,31]. Their molecular structure comprises an electron donor and an electron-deficient ketone group linked to vinyl as an electron acceptor. The D-π-A conformation is responsible for forming the AIEE/AIE feature, which enhances fluorescence in the solid state. In addition, the enone group acts as an electron acceptor and provides the chalcone system with extended π-conjugation, facilitating long-wavelength emissions with large Stokes shifts [11]. These features, with the possibility of using high concentrations without being affected by the detrimental ACQ effect, make the chalcone-based AIE fluorophores compatible with biological systems. The chalcone-based fluorescent probes have been applied to the ratiometric detection of thiols [24], protein [32] and alkaline phosphatase (ALP) [25].

Moreover, taking advantage of the AIE-based biosensors, some fluorophores with AIE characteristics have been designed for the detection and imaging of enzyme activities in the case of different diseases [19,33]. Several chalcone derivatives were synthesised and tested against the AChE enzyme in vitro [34,35,36,37], and the promising results demonstrated the importance of the chalcone scaffold in binding and inhibiting the enzyme. Acetylcholinesterase (AChE) is the enzyme belonging to the family of enzymes responsible for the hydrolysis of choline esters, such as acetylcholine in the nervous system. Inhibition of AChE has been of great interest as a therapeutic strategy for a range of neurological disorders, particularly Alzheimer’s disease (AD). Natural chalcones have been found to possess a small molecular weight and a lower toxicity profile compared to the other AChE inhibitors, making them a more attractive option for the treatment of Alzheimer’s disease [37]. Chalcones and their synthetic analogues hold promise as a potential therapeutic strategy for the treatment of AD, particularly through their ability to inhibit the AChE enzyme and improve neurotransmitter function.

In light of these facts, the initiation of research that will result in finding a potential use for the AIE/AIEE-active natural chalcones seems to be appropriate. The structural and spectroscopic properties of natural chalcones are of continual interest to our team [26,27,38,39,40]. Our attempt consists of evidence that the CA molecules in the aqueous solution exist in the dimeric state, whereas in the ethanol solution, they exist mainly as monomers. Moreover, with the increase in water fraction, the CA molecule shows enhanced emissions induced by the aggregation (AIEE) effect [27].

This paper presents the spectroscopic study of natural chalcones with different substituents in the aromatic rings in the selected solvents, employing UV–vis absorption and fluorescence spectroscopy as well as a structural study using single-crystal X-ray diffraction. Our aim is to investigate, for the first time, the influence of substituents in the aromatic rings on enhanced emission induced by the aggregation in aqueous solutions and in the solid state. Finally, we examine the effect of chalcones with structural diversity, considering 1,1-diphenyl-2-picrylhydrazyl (DPPH) activity as well as acetylcholinesterase (AChE) and butyrylcholinesterase (BuChE) inhibitions to find a relation between the AIEE characteristics and the biological properties of the investigated molecules.

We hope that the data reported here can be a reference for further research on these naturally occurring chalcones, providing clues for understanding their biological and health-promoting properties and helping researchers find guidelines for discovering highly sensitive chalcone-based fluorophores and designing new and efficient materials for potential biological and technological applications.

## 2. Results and Discussion

The versatility of the chalcone skeleton allows for easy modification to create derivatives with varying properties, making it an important area of research in the fields of organic and medicinal chemistry [41,42]. The structure of the chalcones selected for the study of 2′-hydroxy-4,4′,6′-trimethoxychalcone (HCH), cardamonin (CA), xanthohumol (XN), isobavachalcone (IBC) and licochalcone A is presented in Figure 1. The reason for selecting these representatives of the chalcones is their structural diversity, with different substituents in rings A and B, particularly regarding the number and position of the hydroxyl groups. In the structure of HCH, three −OCH_3_ groups and one −OH group are attached to the benzene ring, whereas in the structure of CA, there are one −OCH_3_ group and two −OH groups.

XN and IBC belong to the prenylated chalcones with a hydroxyl group in the 2′, 4′ and 4 positions, and they exhibit diverse and remarkable biological and pharmacological activities. The presence of prenyl groups in the skeleton can affect the lipophilicity, membrane attachment, and transmembrane transport of the molecules [39,43]. Licochalcone A (LIC) is a major phenolic constituent of the liquorice species *Glycyrrhiza inflate*, and it possess a 1,1-dimethyl-2-propenyl group attached to ring B in the chalcone structure [44]. Licochalcone A was chosen as the representative of chalcones without the −OH group in the 2-position of ring A.

### 2.1. Spectral Properties in Solution

The normalised experimental UV–vis spectra of CA, XN, IBC, HCH and LIC in chloroform, methanol, ethylene glycol and H_2_O, with a concentration of 1.0 × 10^−5^ M, are shown in Figure 1. Due to the fact of the low solubility of chalcones in water, appropriate amounts of the samples were dissolved in methanol (stock solution C_0_ = 2 × 10^–3^ M) and then added to H_2_O (~1:99 MeOH:H_2_O *v:v*). The respective locations of the absorption and emission maxima for the solvents used in the study are presented in Appendix A. The UV–vis spectroscopic features are characteristic of chalcones and are due to the strong absorption of the I band at 340–390 nm (the conjugated π system involving the B-ring, cinnamoyl) and the less intense II band located at 240–300 nm (for the A-ring benzoyl system, see Figure 1) [45].

In all studied chalcones, the observed broad band I is characteristic of the π → π* electronic transition in the given chromophoric system. With increasing solvent polarisability (from chloroform up to H_2_O), the effect of the bathochromic shift was observed from 343 to 357 nm for CA, from 365 to 370 nm for HCH, from 367 to 371 nm for IBC, from 366 to 368 nm for XN and from 360 to 380 nm for LIC. The differences in the spectra in aqueous solutions, methanol, ethylene glycol and chloroform were probably caused by both solvent (solvatochromism) and aggregation effects [17]. Our previous paper on cardamonin showed the similarity between the absorption maxima of CA in the aqueous solution and the crystalline state, where two dimeric forms were observed [27]. Moreover, only for licochalcone A, a near mirror image was observed between the absorption and fluorescence emission spectra (Figure 2) in the studied solvents. This similarity is due to the strong π → π* transition between the electronic ground and excited states, with a slight geometry alteration [46].

Due to their conjugated system, chalcones with the proper electron-pulling and electron-pushing functional groups on the benzene rings can be fluorescent, making them potential chemical probes for mechanistic investigations and imaging/diagnosis. As a fluorescent compound, the photophysical parameters, which include the absorption and emission wavelengths, extinction coefficient, and quantum yield (ϕ_F_), are critical for biological applications. The fluorescent intensity of the chalcone-based compounds depends highly on the solvent polarity and the pH, concentration and viscosity of the solution [47]. In the aprotic solvents, for most of the chalcones, fluorescence decreases with decreasing solvent polarity, although the fluorescence is completely lost in the protic solvents, such as water or ethanol, at neutral pH [48,49,50].

For better analysis of the spectroscopic behaviour of all the analysed compounds, we present the emission spectra recorded in the methanol solution with the changing concentration of the sample and with different water–methanol proportions. Figure 2 presents the fluorescence emission spectra (Em) for chalcones in the methanol solution corresponding to the absorbance spectra in Appendix A. The excitation emission was registered at the wavelength of the maximum absorbance and the shoulder located at 300–310 nm (for all concentrations observed in Appendix A). As can be seen in Figure 2, as the concentration of the CA increased, we observed a decrease in the emission intensity in both the shortwave (292 nm) and longwave (342 nm) excitation and the red shift of the band, with the maximum at 373 to 383 nm as well as 378 to 383 nm. When the possibility for hydrogen bonding is great, molecules can combine to form different molecular forms, e.g., crystalline aggregates, dimer-type aggregates, and progressively larger aggregated forms. The emission of fluorescence can be weakened or quenched at the high local concentration associated with the formation of aggregates. On the other hand, for LIC with the increasing concentration, a significant increase in the intensity of the band, with the maximum at 490 nm, was observed. Licochalcone A shows great stability in the methanol solution.

The emission spectra of the tested chalcones in the aggregation state were studied in the MeOH:H_2_O mixtures to investigate the AIEE phenomenon. Figure 3 presents the fluorescence emission spectra of chalcones in the mixed solution MeOH:H_2_O system with different water fractions (*f*_w_). Corresponding absorbance spectra are presented in Appendix A. In the case of CA, the intensity of the emission spectra increases with the increasing content of the water fraction. However, the emission wavelength was red-shifted from 421 to 430 nm when the water fraction changed from 10% to 90%. The observed spectral changes evidence an increase in the number of aggregated forms in the analysed system and the formation of aggregative forms in the respective aqueous systems. The observed effects were similar for HCH (Figure 3e). In the case of prenylated chalcones XN and IBC, as the water fractions increased, the fluorescence intensity changed irregularly. In the aqueous system, the solute molecules can aggregate into different forms, leading to different fluorescence behaviours. A completely different relationship was observed in the case of licochalcone A, wherein, as the water content increased in the mixed solution, the fluorescence intensity decreased gradually and reached its minimum when the volume fraction of water was 10%.

As can be seen, the observed fluorescence effects depend on both the concentration of the compound and the water fraction.

As follows from the literature, RIR can be the main reason for the AIEE phenomenon in the flavonoid derivatives [8]. The free torsional vibrations of the enone moiety and the benzene units in the solution phase result in the energy loss of the excited state through the non-radiative channel, which is responsible for the weak-emissive properties of chalcones. The restricted rotation in the concentrated solutions or cast in the solid system results in the blocking of the non-radiative pathway, leading to enhanced fluorescence intensity [11]. Therefore, in order to study the reason for the AIEE characteristics of chalcones in terms of restrictions to molecular movements, the system viscosities were increased by applying mixed solutions with different ratios of ethylene glycol (EG) in methanol. In Figure 4, when the EG fractions increased from 10% to 90%, the viscosities of the solutions increased gradually, and as a result. the emission intensities of chalcones increased. The appropriate absorbance spectra in the analysed MeOH:EG system are presented in Appendix A. A more tight molecular packing and restrictions to the internal rotation between the three-carbon α,β-unsaturated carbonyl system in the chalcone structure resulted in red-shifted and enhanced emission intensity when compared to their form in methanol. The enone group acts as an electron acceptor and provides the chalcone system with an extended π-conjugation that facilitates long-wavelength emission with large Stokes shifts [51]. It can be speculated that the result of AIEE in chalcones should also be RIR, which is consistent with flavonoids [9].

Notably, LIC and CA molecules exhibited the largest Stokes shifts, ranging from 83 to 134 nm in the investigated solvent, with the fluorescent emission wavelengths ranging from 430 to 515 nm (Appendix A). These wide Stokes shifts are favourable for bioimaging applications because of the reduction of self-quenching originating from molecular self-absorption [10].

#### Fluorescence Quantum Yield

The photophysical properties of chalcones are presented in Table 1. The fluorescent quantum yields (Φ_f_) of CA and HCH in the aqueous solution were higher than those in the other investigated solvents and mixtures (chloroform, MeOH, ethylene glycol and MeOH/water). On the other hand, all compounds also exhibited a large value of Φ_f_ in ethylene glycol. Licochalcone A, being the most sensitive to solvent polarity, has a significantly large value of fluorescent quantum yields in ethylene glycol (50.43%) and in methanol (14.92%). The smallest Φ_f_ values were observed for XN in all investigated systems, which indicates the predominant participation of non-radiative relaxation.

Previously, it was found that chalcones with suitable substituents exhibit desirable fluorescence properties and that this type of chalcone can be investigated as fluorescent probes for many purposes [50,52,53]. Tomasch et al. [54] designed novel fluorescent chalcone-based ligands, exploited in the human histamine H3 receptors (hH3R), to facilitate the visualisation of human hH3R on hH3–HEK cells. Their characteristics showed fluorescence emission maxima larger than 500 nm, which allowed them to avoid the appearance of interference with autofluorescence. This demonstrates the potential of using the intrinsic fluorescence property of chalcones as a multi-tasking tool for biological research that does not require bulky fluorophores, which could have a significant impact on the cellular targets.

Generally, the structure relationship regarding fluorescence properties shows that the molecular planarity of the chalcone moiety plays a critical role in fluorescence [29]. For the A-ring, the weak electron-donating groups are favourable for the quantum yield, but the hydroxy substitution at the 2-position reduces it through the internal hydrogen-bonding interactions with the chalcone ketone moiety. With the exception of LIC, all studied chalcones had the hydroxy group at this position. On the other hand, there are some environmental factors that could affect the fluorescence properties of chalcone-based compounds. Evaluating the solvent effect, we can observe the overall positive correlation between the solvent polarity and the fluorescence quantum yield. As a result of the spectral experiment, licochalcone A has the potential to be used in fluorescence probes, sensors and optoelectronic devices.

### 2.2. Spectral Properties in the Solid State

The restriction via the surrounding polymer chains can also lead to a significant improvement in the luminescence efficiency of chalcones. The crosslinked polymer system would further limit the intramolecular rotation of molecules, which leads to the enhancement of fluorescence intensity. Since chalcone molecules undergo excited-state fluorescence quenching through internal conversion in the solution, thus our next step was to check their photochemical properties in the solid state. Figure 5a–e show the absorption and emission spectra of the investigated chalcones blended in the epoxy resin after the curing process. Firstly, all studied chalcones were dissolved in acetone and then added to the epoxy resin to reach the concentration of 20 mM. The fluorescence intensities of the emission peaks, registered upon excitation close to the main absorption band, are weak in pure acetone (red dash lines) and grow about 10-fold in the solid state (solid red lines). It is likely that during the resin crosslinking, there are polymer interactions with chalcones that efficiently block the rotation of the enone moiety and the benzene units, together with the extended π-π interactions affecting emission intensity. In the case of the pure epoxy resin, the emission peak found at 430 nm could give a small contribution to the spectra of bound chalcones at shorter wavelengths (Figure 5a, dotted black line). LIC and IBC in the epoxy resin show the most red-shifted emission peaks compared to XN, HCH and CA (Figure 5a,d). The emission peak of LIC in the solid state at 513 nm had a position similar to the results for the aggregated forms in the solution (Appendix A). However, it did not change its position upon excitation in the shortwave absorption band (310 nm). In contrast, IBC had a large shift of emission peak of 125 nm relative to the fluorescence maximum emission peak in acetone. The other investigated chalcone–epoxy resin systems exhibited a shift of the maximum emission peak of about 30 nm compared to their solution (Figure 5b,c,e). On the other hand, all compounds displayed large Stokes shifts (84–156 nm; Appendix A) in the epoxy resin, which is a definite advantage for exploring their potential in bioimaging applications. Despite the similar yellow colour in the solution (Figure 5f), chalcones exhibited a more intense colour in the solid state (Figure 5g). The AIEE behaviour of these compounds can also be observed from their photographs under the illumination of UV light at 366 nm. As shown in Figure 5f,g, only CA, LIC, and IBC exhibit the expressed emissions from blue and cyan to green, respectively.

Moreover, chalcone-based fluorophores characterised by the enhancement emission in the solution or the solid state at different wavelengths can be readily converted to the ratiometric response, visible emission colour, enhanced photostability, high sensitivity, and low cytotoxicity benefit live-cell imaging. Dai et al. developed a novel turn-on fluorescent probe based on 2′-hydroxychalcone (HCA), which possesses an excited-state intramolecular proton transfer (ESIPT) coupled with the AIE characteristic of the ratiometric detection of biothiols [24]. Another fluorescent probe based on the phosphorylated chalcone derivative that emitted green colour was applied to detect alkaline phosphatase (ALP), which is considered a biomarker in the diagnosis of many human diseases [25].

### 2.3. SEM

For further study of the AIEE phenomena of CA and HCH in the aqueous system, scanning electron microscopy was used. Figure 6 presents the SEM images of CA (panels a–c) and HCH (panels d–f) particles obtained in the mixed solution MeOH:H_2_O with different water fractions (5:5, 2:8, 1:9 *v:v*). When the water fraction was 50%, CA and HCH appeared patchy and dispersed as slightly spherical particles with an average diameter of about 100 nm. However, with the increase in the content of the water fraction, the particles exhibit different morphologies, forming more prominent and less ordered aggregates.

### 2.4. Single X-ray Diffraction

The crystal structure was determined using single-crystal X-ray diffraction. The collected data were analysed for a better understanding of the relationship between molecular structure and the spectroscopic properties of chalcones. Thus, the attempt was made to obtain crystals of five chalcones CA, HCH, XN, IBC, and LIC from the commonly used solvents, e.g., methanol, ethanol, acetone and tetrahydrofuran (THF) using the slow solvent-evaporation technique. In our previous papers, cardamonin crystals [27] and co-crystals of xanthohumol with pharmaceutically acceptable co-formers were characterised [26]. In this study, good-quality crystals were obtained for HCH. After several crystallisation experiments, crystals of a very inferior quality (R_1_~27%) were obtained for IBC. This is the only known structure of this chalcone, to our best knowledge. In the case of LIC, no crystalline material was obtained.

The obtained data revealed that HCH crystallises in the triclinic space group P-1, whereas IBC and CA crystallise in the monoclinic space group P21/c [27]. The literature reports the available structure of 2′-hydroxy-4,4′,6′-trimethoxychalcone deposited in CDS, which has the monoclinic space group P2_1_/n [55]. In Figure 7, the asymmetric units for HCH, IBC and CA are shown. The data relating to the cell parameters and the final structures are shown in Appendix A. Crystal packing and the related dihedral angles for HCH, IBC and CA are presented in Figure 8 and Appendix A. The crystal structure description of the CA crystal has been presented in the previous paper [27]. Specifications of intermolecular hydrogen bonds observed in the HCH and CA crystals are presented in Table 2.

*HCH crystal*: In the HCH crystal lattice, there are two symmetrically independent molecules (blue and green molecules, see Figure 8a) forming layers along the *c*-axis. Additionally, the layers are alternately organised green and blue HCH molecules, which are related to the symmetry centre. Each HCH molecule can form an intramolecular hydrogen bond between the hydroxyl and carbonyl groups (O-H···O), which is typical of 2′-hydroxychalcone [26,27]. Moreover, in the crystal lattice, two HCH molecules (blue and green species) interact by the weak π···π interactions between rings A and distant 3.594 Å.

*IBC crystal*: Regarding the poor crystal quality of IBC (R_1_~27%), only the analyses of the packing in the crystal structure are possible. In the IBC crystal lattice, the intermolecular hydrogen bond motif between the O3-H3 hydroxyl group and the O1 carbonyl oxygen originating from the neighbouring isobavachalcone molecules form the infinitive chain C(8) (see Appendix A). The ICB molecules form a stack along the *c*-axis and interact through the π···π stacking and van der Walls interactions (Appendix A). Moreover, the IBC molecules form two types of stacks, one related through the glide planes and the other associated with the inversion centre. As in the case of XN, the prenyl groups fill the gaps between the planar parts in the IBC molecules in the crystal lattices (Figure 8c) [26].

Generally, the crystals with a high coplanar conformation will exhibit increased fluorescence emission in the solid state because of the bigger π-electron delocalisation [56]. The angle between the planes of the aromatic rings A and B in the structure of IBC is 17.24°; in HCH, it is 20.14°, whereas in CA, it is 58.05° (Figure 8b,d,f). Moreover, for the xanthohumol co-crystals, rings A and B in XN were rotated with dihedral angles of ~10° [26]. The dihedral angle between the aromatic rings in HCH, IBC and the molecules indicates planarity, like in the analogous structure of flavokavain B [57]. The weak fluorescence of CA, HCH, XN and IBC in the solution can be related to the strong intermolecular π-π staking interactions between the aromatic rings, resulting in decay or relaxation back to the ground state from the excited state via the non-radiative channels. In order to gain more information about the relationship between the fluorescence property and molecular structure, computational methods should be used, as planned in the subsequent paper.

### 2.5. DPPH Activity

Natural chalcones such as cardamonin, isobavachalcone, licochalcone A, xanthohumol and 2′-hydroxy-4,4′,6′-trimethoxychalcone have been found to possess potent antioxidant properties [44,58,59,60]. These compounds can scavenge free radicals effectively, prevent oxidative damage, and improve the overall redox balance in the biological systems.

All tested chalcones (CA, IBC, XN, LIC and HCH) were assessed by the 2,2-diphenyl-1-picrylhydrazyl (DPPH) radical-scavenging activity assay (Table 3). In all the analyses, the data were obtained from three independent experiments and represented as mean ± SD. Antioxidant properties were determined for the samples dissolved in MeOH and DMSO (stock solution C_0_ = 2 × 10^−3^ M). It can be noticed that licochalcone A shows the highest percentage of DPPH inhibition (~29% in DMSO and in MeOH) compared to the other chalcones. As shown in Figure 1, licochalcone A possesses the 4′-OH group in ring A and the 4-OH group, as well as the 1,1-dimethyl-2-propenyl substituent at the 5-position in ring B. Prenylated chalcone XN with three hydroxyl groups (2′-OH, 4′-OH and 4-OH) showed a smaller level of inhibition (~8% in DMSO, ~7% in MeOH), while the remaining samples (HCH, CA and IBC) practically had no scavenging activity (~2%). The HCH chalcone with the three –OCH_3_ methoxy groups, as well as the 2′-OH group, did not show antioxidant properties compared to the other samples (~0.35% in MeOH).

The relationship between the chemical structure of chalcones and their antioxidant properties is complex, and multiple factors can contribute to their activity [44,59]. One key factor is the position of the -OH (hydroxyl) group in the A- or B-ring in the molecular structure of chalcones. Additionally, the number and position of hydroxyl groups can increase the ability of the molecule to donate electrons, further enhancing its antioxidant properties. The studies proved that the presence of the 1,1-dimethyl-2-propenyl substituent at the 5-position in the B-ring (licochalcone A) can also enhance oxidant properties [44]. On the other hand, the presence of more methoxy groups compared to the hydroxyl groups can increase the antioxidant properties [59].

### 2.6. AChE and BuChE Inhibition Evaluation

All the presented chalcones were evaluated by in vitro assay as AChE and BuChE inhibitors. Compared to the neostygmine control, most of the tested chalcones exhibit only a moderate ability to inhibit both cholinesterase enzymes. The results are presented in Table 4. In all the analyses, the data were obtained from three independent experiments, each performed in triplicate (*n* = 9) and represented as the mean ± SD. The values within the study compound are not significantly different (*p* < 0.01) according to the Tukey multiple comparison test.

Previously, a variety of flavonoids were reported to show AChE and BuChE inhibitory activity, including isobavachalcone [61]. The results concerning the acetylcholinesterase (AChE) inhibitory activity showed that licochalcone A is characterised by the most effective activity (IC_50_ 23.41 ± 0.02 µM). On the other hand, for butrylocholinesterase (BuChE), the inhibitory activity of LIC and IBC had the most effective results—IC_50_ 42.28 ± 0.06 µM, IC_50_ 43.05 ± 0.16 µM, respectively. The results indicate that the existence of a free hydroxyl group in both A- and B-rings, as well as the 1,1-dimethyl-2-propenyl group, seems to increase the inhibitory activity of the chalcones. Interestingly, 2′-hydroxy-4,4′,6′-trimethoxychalcone did not show any inhibitory properties towards AChE, whereas in the case of the BuChE enzyme, the activity of the HCH compound was similar to CA (IC_50_ 76.54 ± 3.03 µM and IC_50_ 75.29 ± 0.41 µM). This indicates that, as opposed to acetylcholinesterase, the presence of a methoxy group in the A-ring strongly promotes the inhibition of butyrylcholinesterase activity. Similar conclusions were found by Rosa et al. based on studies of the relationship between the structure and biological activity of hydroxy- and methoxychalcones [59].

Chalcones were found to possess acetylcholinesterase (AChE) and butyrylocholinesterase (BuChE) inhibitory properties, making them promising candidates for the treatment of neurological disorders such as Alzheimer’s disease [61]. On the other hand, further research, for example, on structure–activity relationships (SARs), is needed to fully understand the mechanism of chalcones activity as AChE and BuChE inhibitors and to evaluate their efficacy in human trials [62].

## 3. Materials and Methods

### 3.1. Materials

2′-hydroxy-4,4′,6′-trimethoxychalcone (HCH) with ≥98% purity, cardamonin (CA) with ≥98% purity, xanthohumol from hop (*Humulus lupulus*) (XN) with ≥96% purity, isobavachalcone (IBC) with ≥98% purity and licochalcone A (LIC) with ≥96% purity were purchased from Sigma-Aldrich (Saint Louis, MO, USA). Methanol, ethanol, ethylene glycol, DMSO, acetonitrile, THF and n-hexane were of analytical grade.

### 3.2. Methods

#### 3.2.1. Electronic Absorption and Fluorescence Spectra

The stock solutions of chalcones were prepared by dissolving an appropriate amount of the compound (CA, HCH, LIC, XN, and IBC) in methanol or DMSO in order to obtain the molar concentration 2 × 10^–3^ M. The suitable volume of the solution was added to 2 mL of the given solvent or the mixed solution (MeOH:H_2_O, MeOH:ethylene glycol) in order to obtain the required absorbance intensity. The mixed solution was sonicated for 10 min, and then the U–vis absorption and emission spectra were measured at room temperature.

In the case of epoxy-based sample preparation, the appropriate amount of chalcones dissolved in acetone was added dropwise to 3 g of component A (bisphenol A diglycidyl ether (BADGE)). The mixture was gently homogenised with a mechanical stirrer at room temperature for a few minutes. Then, 1 g of component B (4,4′-methylenebis(2-methylcyclohexylamine) was added to the tube, and stirring was continued for 1 h. The concentration of chalcones in the epoxy resin was 20 mM. The mixtures were poured into the disposable PMMA cuvettes and evaporated for 1 h to remove air bubbles. The samples of epoxy resin blended with chalcones were tested when curing was completed (24 h).

Electronic absorption spectra of chalcones were recorded using the double-beam UV–vis spectrophotometer Cary 300 Bio (Agilent, Santa Clara, CA, USA) equipped with a thermostated tray holder with a 6 × 6 multi-cell Peltier block.

A Cary Eclipse spectrofluorometer (Agilent, Santa Clara, CA, USA) recorded the fluorescence emission spectra. All measurements were made at 22 °C. Both the excitation and the emission slit widths were set at 10 nm. The fluorescence spectra were collected with disposable poly(methyl methacrylate) (PMMA) cuvettes. The scan rate was set at 600 nm/min. The photomultiplier (PMT) voltage ranged between 500 (for the licochalcone-A-based samples) and 600 V (other chalcone-based samples).

Grams/AI 8.0 software (Thermo Electron Corporation; Waltham, MA, USA) was applied for the analysis of the recorded data. The figures were prepared with OriginPro 2021b software (OriginLab Co., Northampton, MA, USA).

#### 3.2.2. Fluorescence Quantum Yield

The quantum yields of chalcone solution fluorescence were determined using 7-diethylamino-4-methylcoumarin (coumarin1) as the fluorescence standard. The measurements were made in ethanol φ_F_ = 0.73 [63]. The final fluorescence quantum yield values were calculated based on Equation (1):(1)$$\Phi_{F\left(X\right)}=\Phi_{R\left(\mathrm{EtOH}\right)}\left(\frac{\lambda_{\mathrm{exR}(\mathrm{EtOH})}}{\lambda_{\mathrm{exX}}}\right)\left(\frac{I_{X}}{I_{R(\mathrm{EtOH})}}\right)\left(\frac{\eta_{X}^{2}}{\eta_{R(\mathrm{EtOH})}^{2}}\right)$$
where the subscript X denotes the corresponding chalcones at different solvents/ratios, $\lambda_{ex}$ is the value of the absorbance at the excitation wavelength, I the plane under the emission curve, and η the refractive index of the solvent. The subscript R denotes the fluorescence standard.

#### 3.2.3. SEM Measurements

Cardamonin (CA) and 2′-hydroxy-4,4′,6′-trimethoxychalcone (HCH) were prepared in the MeOH/H_2_O (5:5, 2:8, 1:9, *v*:*v*) mixtures. The samples were sonicated for 10 min and then added dropwise to the silicon wafers, and the silicon wafers were placed at room temperature for 24 h to evaporate naturally; then, they were measured using the scanning electron microscope. The sample morphology was investigated by scanning electron microscopy with a Zeiss Ultra Plus instrument with Bruker XFlash Detector 5010.

#### 3.2.4. Single X-ray Diffraction

The 2′-hydroxy-4,4′,6′-trimethoxychalcone (HCH), isobavachalcone (IBC) and licochalcone A (LIC) samples were prepared as the previously described for the cardamonin crystals [27]. The samples were first recrystallised from n-hexane-acetonitrile (9:1). The solution was evaporated to obtain plate-like crystals exhibiting a bright yellow colour. These crystals were then dissolved in ethanol, methanol or THF and allowed to stay for 24 h at room temperature.

The single-crystal X-ray diffraction data were collected by means of a Rigaku Oxford Diffraction diffractometer equipped with a MicroMax-007 HF, a rotated Cu anode as the X-ray source (CuKα), multilayer optics and a Pilatus 300 K surface detector at T = 293 K. Data reduction and cell refinement were performed with CRYSALIS^PRO^ [64]. All structures were solved using direct methods [65] and refined using Olex2 software [66]. The refinement was based on the square structure factors (F^2^) for all reflections except those with very negative F^2^ values. Almost all of the hydrogen atoms were in an idealised geometric position except for those forming the hydrogen bonds. Appendix A lists the experimental details for all measured single crystals. The crystallographic data were deposited at the Cambridge Crystallographic Data Center (CCDC; No 2239026 and No 2243539).

#### 3.2.5. DPPH Activity

2,2-Diphenylpicrylhydrazyl (DPPH) was purchased from Sigma-Aldrich (Steinheim, Germany). DPPH radical-scavenging activity was determined according to the method described by Brand-Williams et al. using (+)-catechin (Sigma-Aldrich, Steinheim, Germany) as the positive control [58,67]. The chalcone samples were previously dissolved in DMSO and MeOH at the molar concentration of 2 × 10^–3^ M. The positive control sample was dissolved in DMSO at the same molar concentration. The DPPH stock solution was diluted 10× with methanol to obtain the concentration of 1 × 10^–4^ M. The measurements: 70 μL of the sample was added to 4 mL of the DPPH solution, and the samples were left for 30 or 60 min at room temperature in the dark. The absorbance was recorded at 517 nm by s Varian Cary 50 spectrophotometer. The control was prepared as above without any sample.

DPPH free-radical scavenging activity was calculated using the following Formula (2):(2)$$\mathrm{Scavenging}\;\mathrm{activity}\;[\%]=\frac{\left[\mathrm{Absorbance}\;\mathrm{of}\;\mathrm{control}-\mathrm{Absorbance}\;\mathrm{of}\;\mathrm{test}\;\mathrm{sample}\right]}{\mathrm{Absorbance}\;\mathrm{control}}\times 100$$
where the absorbance of control is the absorbance of DPPH radicals in the absence of compounds, and the absorbance of the test sample is the absorbance in the presence of the studied compound.

#### 3.2.6. AChE and BChE Inhibitor Activities

Acetylcholinesterase (AChE, E.C. 3.1.1.7, from human erythrocytes), butyrylcholinesterase (BuChE, E.C. 3.1.1.8, from equine serum), acetylthiocholine iodide (ATC iodide), butylthiocholine iodide (BTC iodide), 5,5′-dithiobis-(2-nitrobenzoic acid) (DTNB) and neostigmine bromide were purchased from Sigma-Aldrich (Steinheim, Germany). The inhibitory activities against AChE and BuChE of the prepared compounds were performed by means of the method previously developed by Ellman et al. using neostigmine as the reference compound [68]. This is based on the reaction of released thiocholine to give a coloured product with a chromogenic reagent. Seven different concentrations of the compounds in the range 10^−9^–10^−3^ M were measured at 412 nm. All the assays were performed in the 0.1 M NaH_2_PO_4_/Na_2_HPO_4_ buffer (pH = 8) using the Varian Cary 50 spectrophotometer. 5,5′-Dithio-bis(2-nitrobenzoic acid) (DTNB) was used to measure the anti-AChE activity. The test samples and neostygmine were dissolved in dimethylsulfoxide (DMSO) prior to the assay at the stock concentration of 2 mM, and serial dilutions were performed accordingly to obtain the IC_50_ curve. Briefly, 140 µL of 0.1 mM sodium phosphate buffer (pH 8.0), 20 µL of DTNB, 20 µL of the assay solution and 20 µL of the AChE/BuChE solution were added and incubated for 15 min at 25 °C. The reaction was then initiated by adding 10 µL of acetylthiocholine iodide or butylthiocholine iodide. Yellow anion (2-nitro-5-thiobenzoic acid) formation was recorded at 412 nm for 10 min. It was determined by preparing an identical solution of the enzyme in the absence of tested compounds (as the control). Control and inhibitor readings were corrected with a blank read. All operations were repeated three times. The inhibition properties are reported as the IC_50_ values, which were determined graphically from the inhibition curves of log inhibitor concentration vs. the percentage of inhibition. The IC_50_ values represent the concentration of the inhibitor required for 50% inhibition of the enzyme.

## 4. Conclusions

In summary, the spectroscopic and structural features of a series of natural chalcones with different substituents in the aromatic rings were characterised. Their photophysical properties were systematically investigated, revealing that chalcones with a hydroxy substitution at the 2-position in ring A emit weak fluorescence in the protic solvents. Moreover, CA and HCH molecules showed enhanced fluorescent emission in solutions with increasing water fractions. Analysing the experimental results of fluorescence intensity in the solvent system with different viscosities and in the solid state, it was determined that the main reason for the AIEE phenomenon was the restriction of intramolecular rotation (RIR). Single-crystal structure analysis indicates that the studied chalcones (CA, IBC, XN and HCH) possess different crystal systems, space groups, packing structures and conformations but similar crystal morphology, which can influence the enhancement of photoluminescence in the solid state. Licochalcone A, which does not possess a hydroxyl substituent at the 2-position in the A-ring, exhibited high photostability in different solvents, large Stokes shifts (>100 nm) and a high value of fluorescent quantum yields, which make it suitable for biological applications. Moreover, natural chalcones exhibited potent antioxidant properties (DPPH radical-scavenging activity assay) and AChE/BuChE inhibition activity. Regardless of the different AChE inhibition activity of all compounds, the assay identified licochalcone A as the most active compound (AChE IC_50_ 23.41 ± 0.02 μM, BuChE IC_50_ 42.28 ± 0.06 μM). The existence of a free hydroxyl group in both A- and B-rings, as well as the 1,1-dimethyl-2-propenyl group, may increase the inhibitory activity of the chalcones. Putting together the findings stated above, LIC would be a promising candidate as an AIE/AIEE-active chalcone for potential bioimaging applications in living systems.

Finally, this study provides a good example of chalcone-based fluorophores that can be further developed for the synthesis of new compounds with desirable spectroscopic properties for biological applications.

## Data Availability

The datasets used and/or analysed during the current study are available from the corresponding author upon reasonable request.

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
