# Peer review of "Naturally Occurring Chalcones with Aggregation-Induced Emission Enhancement Characteristics"

_molecules, 2023, doi:10.3390/molecules28083412_

Round 1
Reviewer 1 Report
The manuscript "Spectroscopic and structural analyses of natural chalcones showing the aggregation-induced emission enhancement with an assessment of their antioxidant and acetylcholinesterase inhibition potential" is well written and design
only few points are required:
1- The reason for selecting these representatives of the chalcones are missed
2- Absence of a positive control in DPPH assay is questionable
3- There is some typos and syntax
Reviewer 2 Report
The work presented by the Authors is decent, hence I find it compelling and worthy of publication. The only thing I'm personally missing is some kind of quantification of the self-association phenomenon. The Authors have already presented the UV-Vis spectra of studied compounds in the same environment, yet in different concentrations (for instance, Figure S1). In my opinion, such a set of spectra, possibly recorded solely in water or with some addition of other solvents (excluding those, which prevent molecular aggregation, i.e. MeOH and DMSO), could be subjected to chemometric analysis, i.e. principal component analysis (PCA) executed on centered (not autoscaled) spectra. Such analysis could provide some aggregation constants and give a general idea on the concentrations in which the self-association process is being promoted. This could also yield the information on the possible 'stages' of aggregation (monomer -> dimer -> higher aggregate; or maybe the dimerization step is skipped?).
I understand that the Journal gives only ten days to prepare a revised version, so it may not be enough to perform the suggested analysis. Nevertheless, I believe that it would be 'a cherry on the top' of the presented data.
Besides the above, I believe that the general presentation is great, yet I've spotted some spelling mistakes and editorial errors, i.e. page 2, line 70 (double stop); page 5, line 173 (should be 'observed'), line 175 (missing coma at the end); page 8, line 246 (missing space after citation 44); page 13, line 385 ('there should be used' does not feel fluent; maybe 'computational methods should be used' instead? - this construction repeats in the text); page 18, line 569 (missing stop at the end). Also, the sentence on page 2, lines 53-55 is quite difficult to comprehend, I would suggest to revise it.
Reviewer 3 Report
This manuscript titled ‘Spectroscopic and structural analyses of natural chalcones showing the aggregation-induced emission enhancement with an assessment of their antioxidant and acetylcholinesterase inhibition potential’ has the overall goal of characterizing various natural chalcones using a wide array of spectroscopic techniques and structural analyses. The antioxidant and neuroprotective properties of these chalcones were also investigated.
The title is very long, and it is hard to parse the information and what the overall goal of the study is based on the title.
It is hard to appreciate the abstract because it is unclear why the study is needed in the first place. Just reading at the abstract, it looks like numerus analyses were performed but with no prior knowledge on its rationale.
It is unclear if there are related studies already performed for this kind of research. I think it is also best to clarify the novelty of this project in the introduction. In the abstract, it is best to know what the advantages of this type of study are.
AIEE as a phenomenon should perhaps be substantiated and should touch important specific points related to its significance.
In the introduction, it is unclear to me as to what is the relationship between AIE/AIEE effect and the antioxidant, anti-inflammatory, antimicrobial and anticancer properties of these chalcones.
I think in the abstract, it is better to clarify that these chalcones were studies under solutions. Also, it is unclear as to why these chalcones were not extracted from specific sources. In that way, one would be able to assess the yield of chalcones and how much is extracted.
Why is it necessary to study the properties of this chalcone under different solvent conditions?
It looks like there are some sentences in the conclusion that is not aligned with the abstract. For example, the conclusion mentioned ‘potential bioimaging applications in living systems’ but I believe this was not mentioned in the abstract.
I think overall, the manuscript should have a strong rationale and novelty. Also, many aspects mentioned in the manuscript are not consistent with each part. I am not sure how the different spectroscopic and structural analyses align with the assessment of the antioxidant and other properties of the chalcones. It is very hard to appreciate because of this.
